# A Self-Powered Wireless Temperature Sensor Platform for Foot Ulceration Monitoring

**DOI:** 10.3390/s24206567

**Published:** 2024-10-12

**Authors:** Joseph Agyemang Duah, Kye-Shin Lee, Byung-Gyu Kim

**Affiliations:** 1Intel Corporation, Chandler, AZ 85226, USA; joseph.agyemang.duah@intel.com; 2Department of Electrical and Computer Engineering, The University of Akron, Akron, OH 44325, USA; klee3@uakron.edu; 3Division of AI Engineering, Sookmyung Women’s University, Seoul 04310, Republic of Korea

**Keywords:** foot ulceration monitoring, foot temperature sensor, piezoelectric bimorph, self-powered wireless sensor platform

## Abstract

This work describes a self-powered wireless temperature sensor platform that can be used for foot ulceration monitoring for diabetic patients. The proposed self-powered sensor platform consists of a piezoelectric bimorph, a power conditioning circuit, a temperature sensor readout circuit, and a wireless module. The piezoelectric bimorph mounted inside the shoe effectively converts the foot movement into electric energy that can power the entire sensor platform. Furthermore, a sensor platform was designed, considering the energy requirement of 4.826 mJ for transmitting one data packet of 18 bytes. The self-powered sensor platform prototype was evaluated with five test subjects with different weights and foot shapes; the test results show the subjects had to walk an average of 119.6 s to transmit the first data packet and an additional average of 71.2 s to transmit the subsequent data packet. The temperature sensor showed a resolution of 0.1 °C and a sensitivity of 56.7 mV/°C with a power conditioning circuit efficiency of 74.5%.

## 1. Introduction

Recently, there has been an increasing demand for wireless sensor nodes (WSNs) that can monitor physiological changes, social infrastructures, and environmental variations [1,2,3,4,5]. WSNs are realized with sensors that can sense, compute, and transmit data for specific analysis; the advancements made in sensor and wireless technologies have enabled larger WSNs with lower costs in smart devices, which will continue to make our environment greener in the Internet of Things (IoT) era. A critical limitation of the WSN is the energy source that powers the sensor and the wireless module. Traditionally, sensor nodes require batteries as power sources, however, batteries have a limited lifespan and can offer only a finite amount of energy. On the other hand, sensor nodes must work autonomously for a very long time without changing the power source. Moreover, sensor nodes may be in remote areas or on rotating machines where it is difficult to deliver power. Battery maintenance can be expensive and time-consuming, and batteries serve as a pollutant source. Therefore, harvesting ambient energy can be a convenient and less costly alternative to batteries [6]. Point-of-care testing (POCT) devices (which can monitor various health conditions of an individual in real-time at any given location) are becoming prevalent. In particular, the microheater predominantly used in microfluidic-based POCT applications [7] and the wireless self-powered sweat sensor [8] can contribute to advancing healthcare technology and improving quality of life by making POCT simple, always available, and easily accessible.

WSNs and smart objects can replace their batteries with energy harvesters. Sunlight, vibration, RF, and thermal energies are some available energy sources that can be scavenged. Piezoelectric material acting as a transducer to convert ambient vibration energy to useful electrical energy is one of the best options for self-powered WSNs. Piezoelectric transducers can also convert mechanical strain, such as human movement, into electric power [9]. Examples of self-powered WSNs that use a piezoelectric transducer as an energy source are presented in [10,11,12,13], and include gas pipeline monitoring, wind speed sensing, wildfire detection, and traffic management applications.

Foot temperature monitoring is one of the most effective methods to detect inflammation that leads to ulceration (in diabetic patients experiencing foot complications) [14]. Elevated foot temperature can alert the patient to seek treatment or about reduced activity until the temperature is normalized, which can effectively prevent foot ulcerations. However, although temperature monitoring can be easily performed, periodic monitoring can be cumbersome and time-consuming for the patient. As such, a self-powered wireless sensor mounted on the shoe, which can continuously monitor foot temperature and transmit sensor data to the readout device, will contribute to ulceration prevention for diabetic patients. Existing work related to self-powered WSNs presented in [15] shows the feasibility of using foot movement as the energy source to power an RFID tag. The wireless sensor platform presented in [16] shows a 6 × 9 mm^2^-sized MEMS piezoelectric transducer that can power a Bluetooth low energy (BLE) module and send sensor data to the smartphone. A multi-layered polyvinylidene fluoride (PVDF)-based shoe energy harvester proposed in [17] can generate an average energy of 0.25 mJ for every 2~3 steps.

The proposed self-powered wireless temperature sensor platform combines the piezoelectric bimorph, power conditioning circuit, temperature sensor readout circuit, and BLE module, and can continuously monitor foot temperature and transmit sensor data to the external readout device, where the PC application can display, store, and transmit the temperature information. The prototype sensor platform was mounted on a shoe, where the performance was evaluated in real real-life setting. The novelty of the proposed work is the ability to monitor the foot temperature while the patient is walking, which is a routine activity, and transmit the temperature data to the base station via the BLE module to track and store the foot temperature. Moreover, the sensor platform is energized by the piezoelectric bimorph, which can effectively generate power from foot movement due to walking.

The remaining portion of the paper is organized as follows. Section 2 and Section 3 describe the hardware and software modules of the proposed self-powered sensor platform. The prototype sensor platform test results are addressed in Section 4, and finally, the conclusions are given in Section 5.

## 2. Sensor Platform Hardware

The hardware module of the proposed sensor platform includes the piezoelectric bimorph, which extracts energy from foot movement, a power conditioning circuit that converts the extracted energy into electric power, a temperature sensor readout circuit, and a BLE device. Figure 1 shows the hardware block diagram of the sensor platform.

### 2.1. Piezoelectric Bimorph

Piezoelectric materials produce an electric charge when subjected to mechanical pressure or strain, which is called the piezoelectric direct effect. Piezoelectric transducers typically work in either 33-mode or 31-mode [18]. In 33-mode, a force is applied in the same direction as the poling direction (i.e., the direction in which the electrical dipoles are aligned), involving compression of the piezoelectric solid with electrodes on its top and bottom surfaces. On the other hand, in 31-mode, a lateral force is applied in the direction perpendicular to the poling direction. For foot energy harvesting, two main phases of walking motion (as shown in Figure 2) can be used to generate power [19]. The *contact phase* occurs when the heel strikes the ground, and the *propulsive phase* occurs when the foot bends after landing. For the proposed self-powered sensor, the contact phase is used, since this phase can generate more power than the propulsive phase due to higher excitation in 31-mode and higher mechanical force when the heel hits the ground compared to when the foot bends. The high force that is exerted on the piezoelectric material by the heel presents a bit of a conundrum. On the one hand, we want to have the highest amount of mechanical compression on the piezoelectric material as possible, and on the other hand, too much pressure will cause the piezoelectric generator to lose its curved nature and break. As such, the two factors are considered for building the mechanical structure of the piezoelectric bimorph.

The piezoelectric bimorph was formed by connecting two 3-inch by 2-inch piezoelectric generators in parallel, which decreased the optimum load for maximum power transfer, and increased the amount of available energy. A commercial piezoelectric generator, Thunder TH-6R (Face Companies, Norfolk, VA, USA) [20], which is a multi-layered ferroelectric device that consists of stainless steel, aluminum, and lead zirconate titanate (PZT) superstrated with epoxy, was used to build the bimorph. Furthermore, to measure the DC output power, a diode bridge rectifier *D*_1_~*D*_4_, a storage capacitor *C*_1_, and road resistor *R*_1_ were connected to the piezoelectric bimorph, as shown in Figure 3a. The value of *C*_1_ is chosen so that the time constant *R*_1_·*C*_1_ is greater than the compression and relaxation period of the bimorph. After mounting the bimorph on a shoe, the foot was raised and lowered to the ground with a frequency of one step per second to simulate the walking motion with two phases. Figure 3b shows the output power generated from the bimorph with different *R*_1_ values, where the maximum output power of 1.2 mW is obtained with *R*_1_ = 1.6 MΩ. As a result, with the optimum *R*_1_, the energy generated for each step with a walking period of 1 s is 1.2 mJ, which is obtained from the following relationship:(1)$$E=P\cdot T_{w}\;\;\;\;$$
where *P* is the output power given as *P* = *V*^2^/*R*_1_ and *T_w_* is the walking period, which is set to 1 s.

### 2.2. Power Conditioning Circuit

Figure 4 shows the power conditioning circuit, which includes the diode bridge rectifier *D*_1_~*D*_4_, storage capacitor *C*_2_, load resistor *R*_1_, PMOS *Q*_1_, NMOS *Q*_2_, and voltage regulator LT3060 (Texas Instruments, Dallas, TX, USA), where Schottky diodes (*D*_1_–*D*_4_) with a forward voltage drop of 0.4 V are used to minimize the diode forward voltage drop. The electric charge from the rectified piezoelectric bimorph is stored in the storage capacitor *C*_2_. In addition, *Q*_1_ and *Q*_2_ function as voltage-level-controlled switches that control the release of accumulated energy from *C*_2_. The operation of the circuit is divided into two stages. In the charging stage, the storage capacitor charges to the preset value, and in the discharge stage, it discharges to power the rest of the circuit. During the charging stage, *Q*_1_ is off, and the voltage regulator is disconnected from *C*_2_, *D*_1_–*D*_5_, *R*_1_–*R*_4_, *Q*_2,_ and the piezoelectric bimorph. When *C*_2_ charges to *V_H_*—the preset upper bound voltage across *C*_2_—*Q*_2_ turns on, where *V_H_* is determined by *D*_5_, a 10 V Zener diode, and the resistor divider formed by *R*_2_ and *R*_3_. During the charging of *C*_2_, *D*_5_ prevents the current from sinking into the gate of the *Q*_2_. When the voltage across *C*_2_ reaches 10 V, *D*_5_ starts to conduct, thus the current sinks into *R*_2_, *R*_3,_ and the gate of *Q*_2_. When the voltage across *R*_3_ exceeds 2.2 V, the threshold voltage *V_thn_* of the NMOS, *Q*_2_ turns on. Since there is a current path to ground for *R*_1_ when the voltage drop across *R*_1_ exceeds −1 V—the threshold voltage *V_thp_* of the PMOS—*Q*_1_ turns on. When *Q*_1_ turns on, *C*_2_ discharges through *Q*_1_ and the voltage regulator to the rest of the circuitry. Immediately after *C*_2_ starts discharging, *R*_4_ (4.4 MΩ) forms positive feedback to guarantee that *Q*_1_ is on, which is enabled by forming another voltage divider with *R*_3_. The voltage regulator generates an output voltage of 2.5 V with an input voltage range from 1.6 V to 45 V and a maximum output current of 100 mA. This is ideal for the proposed power conditioning circuit since *C*_2_ starts to discharge (turns on) through the voltage regulator when the voltage across *C*_2_ reaches *V_H_*.

During the discharge stage, the Zener diode turns off when *C*_2_ starts to discharge, meaning *R*_2_ is floating. As a result, the voltage divider formed between *R*_4_ and *R*_3_ now controls the gate voltage of *Q*_2_. When the voltage across *C*_2_ reaches *V_L_*, the voltage across *R*_3_ becomes less than *V_thn_*, and *Q*_2_ turns off. This also turns off *Q*_1_, since there is no current path to ground for R_1_ anymore. The voltage regulator and the rest of the circuitry are disconnected from *C*_2_ again. Capacitor *C*_4_ is a bypass capacitor for the voltage regulator. Since the voltage across *R*_3_ determines the charging and discharging process, the values of *R*_2_, *R*_3_, and *R*_4_ can be obtained using the voltage divider formula. The node voltages during the charging stage are given as follows:(2)$$\left(\frac{R_{3}}{R_{2}+R_{3}}\right)\cdot \left(V_{H}-V_{k}\right)=V_{thn}\;\;\;\;$$
where *V_k_* indicates the Zener diode voltage. Moreover, the node voltages in the discharging stage are described as follows:(3)$$\left(\frac{R_{3}}{R_{3}+R_{4}}\right)\cdot V_{L}=V_{thn}-0.1\;\;\;$$
where the voltage across *R*_3_ is set to 0.1 V lower than *V_thn_* to turn off *Q*_2_. By solving Equations (2) and (3), with values *V_H_* = 12.6 V, *V_L_* = 4.1 V, *V_k_* = 10 V, and *V_thn_* = 2.2 V, the relationship between *R*_1_, *R*_3_ and *R*_3_, *R*_4_ are given as *R*_3_ = 5.5·*R*_1_ and *R*_4_ = 0.95·*R*_3_. Using standard resistors, the values of *R*_2_, *R*_3_, and *R*_4_ are 0.8 MΩ, 4.42 MΩ, and 4.12 MΩ, respectively. However, *R*_1_ is set to 22 MΩ, since *R*_1_ must be large enough, such that when *Q*_1_ turns on, immediately, *Q*_2_ turns on. In addition, *C*_2_ should be large enough to provide energy for the sensor platform. As such, the minimum value of *C*_2_ is determined by the minimum energy required by the sensor platform, which is experimentally determined to be 4.826 mJ per transmission cycle (in Section 4.1). The amount of energy extracted from *C*_2_ between *V_L_* and *V_H_* is given as follows:(4)$$\Delta E=\frac{C_{2}\cdot {\left(V_{H}-V_{L}\right)}^{2}}{2}\;\;\;\;\;$$
where the value of *C*_2_ can be extracted by setting Δ*E* = 4.826 mJ, which leads to 133 uF.

Figure 5a,b show the voltage waveform across *C*_2_ and the voltage regulator output waveform obtained from the circuit simulator. The red waveform shows the voltage across *C*_2_, and the green waveform shows the voltage regulator output. The voltage across *C*_2_ rises until it reaches *V_H_* (=12.6 V) and drops until it reaches *V_L_* (=4.1 V). The charge accumulated in *C*_2_ during the charging process (0 V to *V_H_*) is transferred during the discharge process (from *V_H_* to *V_L_*) through the voltage regulator. Figure 5b shows the close-up view of the voltage regulator output and *C*_2_ voltage transition from 12.6 V to 4.1 V.

As described above, the proposed power conditioning circuit is simple, since it is based on the gate voltages of the transistors to determine the upper and lower bound voltages, *V_H_* and *V_L_*. This enables precise and fast switching during the turning on and off process. Also, using the power rail instead of the ground as in [15,16] to separate the voltage regulator from the remaining circuitry allows fast switching. Furthermore, the proposed power conditioning circuit is realized with fewer components than [15], thus leading to lower power loss during operation.

### 2.3. Temperature Sensor Readout Circuit

The temperature sensor used for the proposed self-powered sensor platform is a resistive-type transducer made from multi-walled carbon nanotubes (MWCNTs) and polypyrrole (PPy) with electro-spun nylon-6 [21]. The resistance corresponding to the temperature is converted into a voltage, which is applied as the input for the analog-to-digital converter (ADC) in the BLE module. The sensor shows a difference in resistance of about 300 Ω within the temperature range from 25 °C to 45 °C, which is an average temperature range in a shoe. This means that a 1.5 Ω change in resistance corresponds to a 1 °C change in temperature. The resistance variation is converted into a detectable voltage through a simple resistor bridge circuit and single supply instrumentation amplifier INA122P [22], which is a high-precision, high-gain, and low-offset amplifier. Figure 6 shows the complete temperature sensor readout circuit, where the voltage across the sensor *R_S_* is set so that it does not exceed 0.8 V since polypyrrole starts to oxidize at voltages higher than 0.8 V [21].

The transfer function of the readout circuit is given as follows:(5)$$V_{out}=\left(V_{in+}-V_{in-}\right)\cdot \left(5+\frac{200k}{R_{g}}\right)\;\;$$
(6a)$$V_{in+}=\left(\frac{R_{P}}{R_{P}+R_{1}}\right)\cdot V_{s}\;\;$$
(6b)$$V_{in-}=\left(\frac{R_{S}}{R_{S}+R_{1}}\right)\cdot V_{s}\;\;$$
where *R*_1_ and *R*_2_ are fixed resistors, *R_P_* is the potentiometer, *R_S_* is the sensor resistance, *R_g_* is the gain resistance, and *V*_s_ is the supply voltage. Considering the value of *R_S_* at 25 °C, and using Equations (5) and (6a,b), the value of *R*_1_ is set to 15 kΩ. This will make the input nodes *V_in_*_+_ and *V_in_*_−_ 2.5 V at 25 °C. Furthermore, in order to make the range of *V_out_* between 0 V to 1.25 V for a temperature range of 25 °C to 45 °C, *R_g_* is set to 2.7 kΩ.

Before performing the temperature measurement (experiment to determine the transfer function between the temperature and output voltage of the temperature sensor’s readout circuit), it is necessary to calibrate the resistive temperature sensor used in this work. This is important because the initial resistance can change with environmental conditions, such as humidity, and with manufacturing variations due to the conditions under which they were made. Toward this end, we measure the initial resistance of the sensor at 25 °C (room temperature). We set the initial resistance of the potentiometer to a value higher than the sensor’s resistance so that the output voltage of the amplifier is about 100 mV, and then start the temperature sensing. The sensor calibration is a one-time process that is performed before using the sensor platform, where the calibration compensates for the manufacturing variations of *R*_1_ and *R_p_* used in the temperature sensor readout circuit, as shown in Figure 6. To find the output voltage of the temperature sensor readout circuit, the temperature was varied from 25 °C to 45 °C (target detection range), and the output voltage corresponding to each temperature level was monitored for 100 cycles, where the average temperature–voltage relationship is given as follows:(7)$$T=17.63\cdot V_{out}+23.40\;\left[{}^{\circ}C\right]$$

Based on Equation (7), the sensitivity Δ*V*/Δ*T* of the proposed temperature sensor is 56.7 mV/°C.

### 2.4. BLE Module

The BLE module used in this work is a radio frequency transmitter BR-LE4.0-S2A (BlueRadios, Denver, CO, USA), which is a proprietary version of CC2540 SoC [23]. The rectangular antenna embedded in the module helps to reduce board space and the number of components. The main functions of the BLE module are to convert the temperature sensor output into a digital format, make connections with the Bluetooth dongle at the base station, and transmit the temperature sensor data.

## 3. Sensor Platform Software

The primary function of the software (ver. 1.0) in this work is to transmit physiological data from the sensor module to the BLE dongle at the base station. To do this, the software performs functions such as scanning for BLE modules, establishing a connection, sending a request, receiving sensor data, disconnecting, and displaying received data. Figure 7 shows the software block diagram of the sensor platform.

### 3.1. Embedded Code Structure

The embedded code for the BLE module is realized by using the application program interface (API) that enables the creation of custom applications. Figure 8 shows the various stages of the BLE module’s embedded code and how it is used to transmit data packets to the Bluetooth dongle connected to the PC. The BLE module powers up when it receives power from the voltage regulator. During power-up, the module initializes all the hardware blocks, powers-up notifications, as well as sets up events, generic attribute profile (GATT) services, and user applications. The BLE module then waits to receive over-the-air (OTA) BlueRadios serial port (BRSP) mode connection requests. Once the BLE module receives a BRSP mode connection request, it connects to the Bluetooth dongle. After a connection is established, the sensor data request packet is sent from the Bluetooth dongle to the BLE module. The packet is received through the UART of the BLE module. If the BLE module does not receive a sensor data request packet, it continues to wait for the data request packet. The Bluetooth dongle continuously transmits a sensor request packet until it receives a packet of data. The data packet together with a CRC is built and transmitted to the Bluetooth dongle after the BLE module receives the request for sensor data.

### 3.2. Data Protocol

The data protocol involves a set of digital rules for data exchange between the BLE module and the Bluetooth dongle. The protocol consists of the header and the payload, where the header is divided into the packet type and a CRC polynomial for data integrity checking. The payload includes a sequence number, different data packets, and a flag that indicates the end of the payload. For the proposed self-powered wireless sensor platform, a stop-and-wait automatic repeat request (ARQ), as described in Figure 9, is implemented. In this case, the BLE module sends the first packet with a zero-sequence number (S0). Upon receiving the packet, the base station (dongle) checks the sequence number for zero and then sends an acknowledgment (ack1) in the form of a packet for the following data. The BLE module then transmits the next packet with a sequence number of one (S1). The process repeats with the base station sending an acknowledgment (ack0) for the next data packet with the sequence number zero (S0). In addition, the timeout is included at both ends to ensure the safe delivery of data. As shown in Figure 9, when the BLE module does not receive an acknowledgment (example ack1) within a specific time window (T0), it re-transmits the packet with the same sequence number (S0) as the previous transmission. The base station also waits for a specific time window (T0) to receive data (for example, with the sequence number, S1) from the embedded module, and then sends another acknowledgment (ack1) for data if it did not receive any data within the specified time frame. The base station always asks for the current data and does not care about previously sampled data, as data from the proposed sensor platform (regarding foot temperature) do not vary much between sampling times. The foot temperature measurement is performed with an accuracy of 0.1 °C using a 16-bit data size, where the packet size is limited to 18 bytes.

### 3.3. PC Application

On the base station side, the PC application should scan for BLE modules, establish a connection with the BLE module, and receive the temperature sensor data. In addition, storing and displaying the sensor data are realized through a graphical user interface (GUI), where the code structure uses events for executing tasks. Once the Windows form loads, the BLE dongle opens the COM port connected to the BLE device, scans for BLE devices, sets up the event for the BLE dongle to receive commands, and registers to receive incoming data packets. For secure data transmission and to protect patient privacy, BLE security level-3, which supports authenticated pairing with encryption, can be used, where the BLE module encrypts its data transmission with a stream cipher [23]. As a possible alert mechanism for elevated foot temperature, the base station can activate an alarm, and send an email or text message to the patient that will provide treatment procedures.

## 4. Experimental Results

### 4.1. Sensor Platform Power Requirement

In order to find the amount of energy required for the sensor platform to transmit one data packet, the average current and duration of each operation stage are measured. The required energy for the data packet transmission cycle will determine the amount of energy that the piezoelectric bimorph should generate to power the sensor platform, and consequently, this will specify the minimum walking time of the subject to generate enough energy to transmit a data packet to the base station.

Figure 10 shows the setup for measuring the power consumption of the sensor platform, which includes the DC power supply (2.5 V), oscilloscope, 10 Ω resistor, sensor module, and BLE dongle. The power consumption will be obtained by measuring the current drawn by the sensor module during various operation stages starting, from power-on to data transmission. To measure the current drawn for each operation stage, the sensor module is connected to the 2.5 V power supply through a 10 Ω resistor, where the average current flowing through the 10 Ω resistor is obtained from the voltage waveform across the resistor. In this case, the BLE dongle and sensor module are pre-programmed with BlueRadios AT commands [23] and custom codes, where the module’s advertising duration is configured so that there is only one advertisement when the module is powered. In addition, the BLE dongle is configured so that only one channel is used for advertisement. The connection event parameters are also changed to limit the number of connections to one when the module is connected to the dongle during power-up. The current drawn by the sensor module during packet transmission stages, which include power-up, advertising, ADC sampling, data received, and transmission through the UART, is measured. However, since the current drawn by each component will be different, the average current will be used. Thus, we have the following:(8)$$Average\;Current=\frac{T_{S}\times I_{C}}{T_{t}}$$
where *T_S_* is the on duration of each component when drawing the current, *I_C_* is the current drawn by each event, and *T_t_* is the duration of each event.

Figure 11 shows the power consumption and duration of each operation stage obtained from the current, where the total power consumption for one transmission cycle is 24.98 mW with a duration of 190 ms. As a result, the BLE module consumes an energy of 4.746 mJ per data packet transmission. In addition, noticing that the sensor readout circuit (sensor bridge circuit plus instrumentation amplifier) consumes 0.08 mJ, the total energy consumption of the sensor platform is 4.826 mJ.

### 4.2. Sensor Platform Test Results

The self-powered wireless temperature sensor platform prototype was realized by placing the piezoelectric bimorph inside the shoe under the insole and mounting the PCB—including the power conditioning circuit, temperature sensor readout circuit, and BLE module—on the shoe. Figure 12 shows the piezoelectric bimorph placed in the shoe and the PCB photo. However, the storage capacitor, *C*_2_ of 330 uF, was used instead of the calculated value of 133 uF, considering the current leakage and the equivalent series resistance (ESR) of electrolytic capacitors. The temperature sensor was also placed in the shoe, where the sensor was in close contact with the foot. In this case, the potentiometer in the resistor bridge circuit was tuned so that the output of the instrumentation amplifier was set to 100 mV. Once the shoe was worn, the subject started walking at a rate of one step per second. A size-10 (US standard) sport shoe was selected for sensor platform testing since this is the most general and comfortable shoe type that people wear. However, we expect the sensor will work properly with different shoe types, excluding high heels and flip-flops. In addition, the test was performed on an indoor flat floor to eliminate other environmental effects. Prior to testing the sensor platform with test subjects, the piezoelectric bimorph was evaluated by a mechanical shaker; the bimorph was stimulated with an average walking frequency (0.5 Hz) and force equivalent to weights of 40 kg to 100 kg, where the piezoelectric bimorph properly generated power with the given force. Thus, we expect the bimorph will properly operate with patient weights from 40 kg to 100 kg. The total cost of building the prototype sensor platform is around USD 295, which includes two Thunder piezoelectric actuators, TH-6R (each USD 100), a BLE module (USD 10), other electric components (USD 5), and PCB fabrication (USD 80). However, considering that component prices are reduced for larger quantity orders, in particular, TH-6R is reduced to USD 10 for 100+ orders, and the manufacturing cost for the actual product will be less than USD 50.

Figure 13a shows the voltage across *C*_2_ (blue waveform), where τ_1_ and τ_2_ are the charging times of *C*_2_ from 0 V to *V_H_* and from *V_L_* to *V_H_*, respectively. As shown, τ_1_ takes 115 s for *C*_2_ to charge from 0 V to *V_H_* and τ_2_ takes 70 s for *C*_2_ to charge from *V_L_* to *V_H_*. This means that a subject needs to walk 115 s to generate enough power to transmit one data packet, and an additional 70 s of walking is required to generate power to transmit the next data packet. In addition, Figure 13b shows the close-up view of the *C*_2_ voltage changing from *V_H_* to *V_L_* and the voltage regulator output, where the discharge time of *C*_2_ is around 350 ms.

Once the connect request at the base station has been activated, the Bluetooth dongle continuously attempts to connect to the BLE module. During the discharge period of *C*_2_, the PC application in the base station connects to the BLE module, where the BLE module samples the temperature sensor data from the instrumentation amplifier output, converts the data into digital temperature data format, and transmits the data over the air to the PC application. The foot temperature is displayed by the PC application through the GUI, where the temperature data are updated each time the PC application receives new data from the BLE module. Figure 14 shows the PC application (GUI) that displays the temperature recording, where the real-time display on the GUI is a result of 11 temperature readings performed within 15 min. In addition, the left side of the GUI shows the most recent temperature reading and the graph on the right side shows the accumulated data. The control buttons and the embedded device address are also shown in the top right corner.

We verified the operation of the proposed sensor platform on five subjects with different foot sizes, shapes, and weights, where the piezoelectric bimorph properly functioned for all subjects. Table 1 shows the test results, where *T*_1_ and *T*_2_ indicate the walking times required to generate power to transmit the first and subsequent data packets, respectively. The min and max values for *T*_1_ and *T*_2_ show 113 s to 126 s and 67 s to 74 s with average *T*_1_ and *T*_2_ times of 119.6 s and 71.2 s, implying that the foot size and weight of the subject do not critically affect the operation. As such, although we were not able to test the sensor platform with subjects that had abnormal foot conditions such as edema or poor blood circulation, we expect that foot conditions will not critically affect the operation as long as the subject has the proper contact and propulsion phase in their walking pattern. Furthermore, since the bimorph with a size of 8 cm × 5.5 cm and thickness of 30 mm can be easily added to the shoe (under the insole), it did not result in any discomfort, gait pattern change, or potential risk of injury to the test subjects. Even though the patient is less mobile and has irregular walking patterns, as long as the patient has proper foot movement with proper contact, midstance, and propulsion phase, the piezoelectric bimorph will still generate power. However, if the patient has slow movement, it will take longer to generate the power required for sensor data transmission. For patients who are immobile or unable to walk for a certain period, a battery can be used to power the sensor platform. With a regular 3.3 V coin cell battery, the BLE module (the most power-hungry component) can operate for up to one month. The durability test of the piezoelectric bimorph has not been performed; however, based on the Thunder TH-6R datasheet [20], the bimorph can withstand 10^8^ operation cycles. Thus, considering an average walking step takes about 2 s, the bimorph can continuously operate for up to 4.5 years. Although the sensor platform has not been tested under different environmental conditions, based on the Thunder TH-6R datasheet [20], the piezoelectric bimorph can be operated within the temperature range of −25 °C to 80 °C and humidity range of 10% to 90%.

### 4.3. Power Conditioning Circuit Efficiency

The efficiency of the power conditioning circuit can be calculated from the energy stored in the storage capacitor, *C*_2_, and the discharge interval measured in Figure 13b, which is given as follows:(9)$$Efficiency=\frac{E_{B}/D_{B}}{E_{sc}/D_{sc}}\times 100\;\left(\%\right)$$
where *E_B_* is the experimentally measured energy required to transmit one data packet (4.826 mJ), *D_B_* is the duration for transmitting one data packet (190 ms), *E_sc_* is the energy stored in the storage capacitor (11.92 mJ), and *D_sc_* is the discharge interval of the storage capacitor (350 ms). Based on Equation (4), the energy stored in the storage capacitor, assuming it discharges from *V_H_* to *V_L_* for the capacitor value of 330 uF, is given as 11.92 mJ, which leads to an efficiency of 74.5%.

### 4.4. Comparison with State-of-the-Art Foot Temperature Sensors

Table 2 shows the major features of the proposed work compared to state-of-the-art foot temperature sensors mainly used for diabetic foot detection. In [24], the foot temperature is monitored using an IR camera (FLIR LEPTON, Wilsonville, USA), where the camera image is sent to the microcontroller that generates the foot temperature map. The temperature map can also be sent to the healthcare provider’s web server via the BLE module for further diagnosis. The temperature sensor proposed in [25] uses non-contact IR temperature sensors to monitor the foot temperature, where the sensors are placed inside a custom insole that can be inserted into the shoe. In this work, the sensor data are sent to the cloud server via a Wi-Fi module. A laser-induced graphene (LIG)-based foot temperature sensor is proposed in [26], where the microcontroller with a BLE module can send the sensor data to the base station that generates the foot temperature map. As shown, the state-of-the-art foot temperature sensors use an IR camera, non-contact IR, and LIG to detect the foot temperature, and can transmit the sensor data via a wireless module. However, existing works use batteries to power the sensor, whereas the proposed scheme uses an energy harvester to energize the sensor platform.

## 5. Conclusions

In this work, a self-powered wireless temperature sensor platform that can be used for real-time foot ulceration monitoring for diabetic patients is proposed. The proposed self-powered sensor platform consists of a piezoelectric bimorph, a power conditioning circuit, a temperature sensor readout circuit, and a wireless module. The piezoelectric bimorph mounted inside the shoe effectively converts the mechanical stress due to the compression and relaxation of the foot movement into electric energy, which can power the entire sensor platform. The power conditioning circuit controls the charging and discharging stages of the piezoelectric bimorph to extract maximum energy from the shoe. Furthermore, the wireless module is realized with a Bluetooth low-energy device for temperature sensor data transmission, where the sensor platform is designed, considering the energy requirement of 4.826 mJ for transmitting one data packet. The self-powered sensor platform prototype was evaluated with five test subjects with different weights and foot shapes; the test results showed that the subjects had to walk an average of 119.6 s to transmit the first data packet and an additional average of 71.2 s to transmit the subsequent data packet. The temperature sensor showed a resolution of 0.1 °C and a sensitivity of 56.7 mV/°C with a power conditioning circuit efficiency of 74.5%.

The challenges of the proposed work include effectively designing the piezoelectric bimorph so that it does not cause discomfort to the patient, maximizing the efficiency of the power conditioning circuit, and obtaining stable characteristics for the MWCNT temperature sensor. The main limitation involves the required walking time to generate enough power to transmit a data packet. As part of future work, once the piezoelectric bimorph failure is detected, the sensor platform can be powered by a battery that is also a backup energy source for immobile patients. For Bluetooth connection failure, in case there is no response when the base station requests data from the sensor, the base station can send an alert to the patient to inform them of the connection failure. Incorporating multiple temperature sensors at different locations of the shoe can be considered so that a temperature profile of the foot can be generated. Other applications of the proposed sensor platform include tracking treatment results for foot ulcers and general POCT devices that benefit from real-time foot temperature measurements.

## Figures and Tables

**Figure 1 sensors-24-06567-f001:**
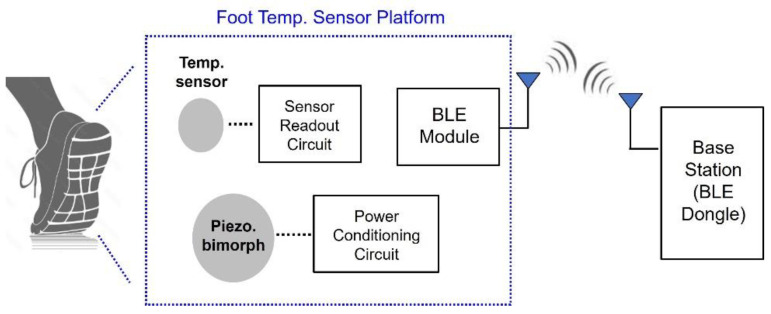
Sensor platform hardware block diagram.

**Figure 2 sensors-24-06567-f002:**
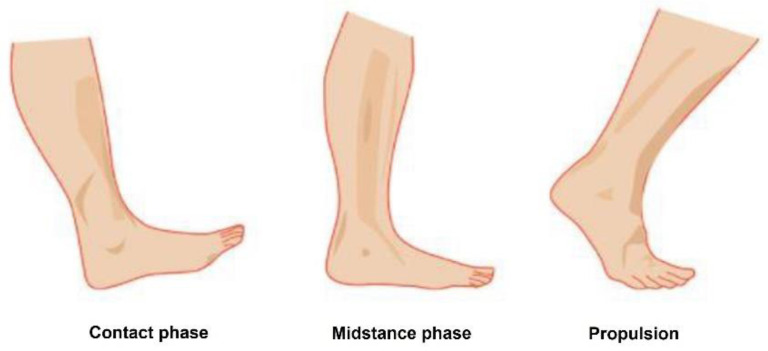
Movement of the foot, divided into two phases.

**Figure 3 sensors-24-06567-f003:**
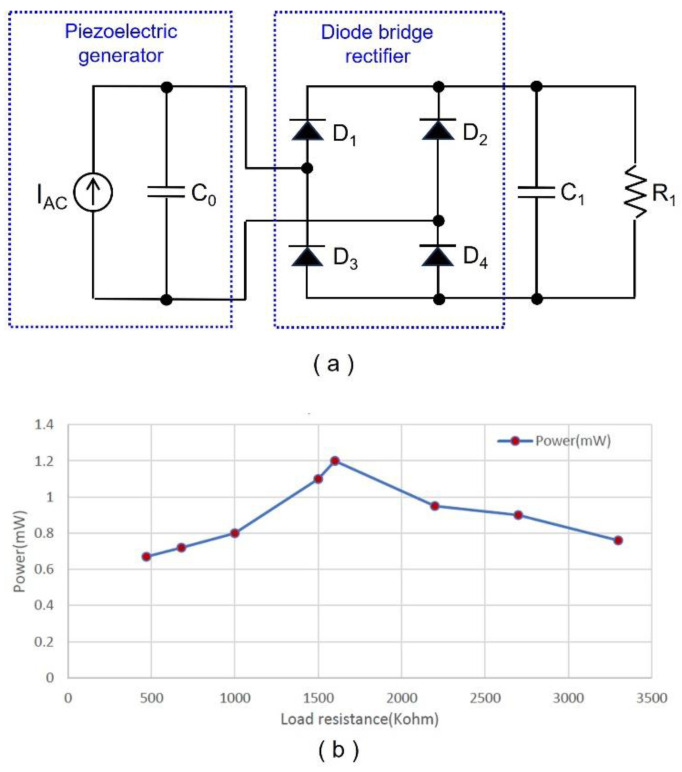
(**a**) Bridge rectifier and load resistance connected to the piezoelectric generator. (**b**) Output power vs. load resistance.

**Figure 4 sensors-24-06567-f004:**
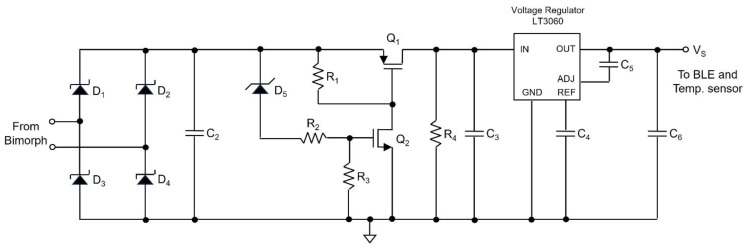
Power conditioning circuit.

**Figure 5 sensors-24-06567-f005:**
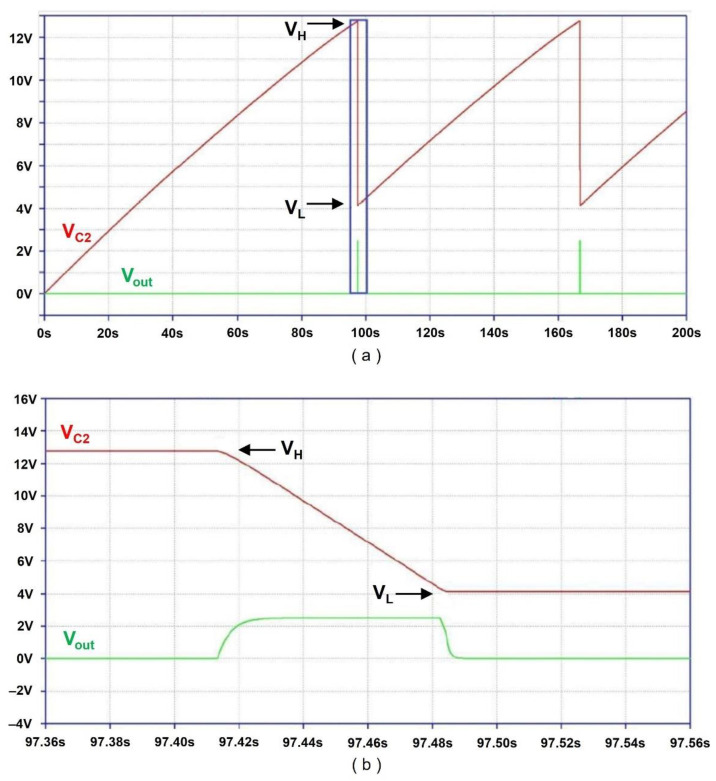
Simulation waveforms (**a**) *C*_2_ voltage and regulator output. (**b**) Close-up view of *C*_2_ voltage and regulator output.

**Figure 6 sensors-24-06567-f006:**
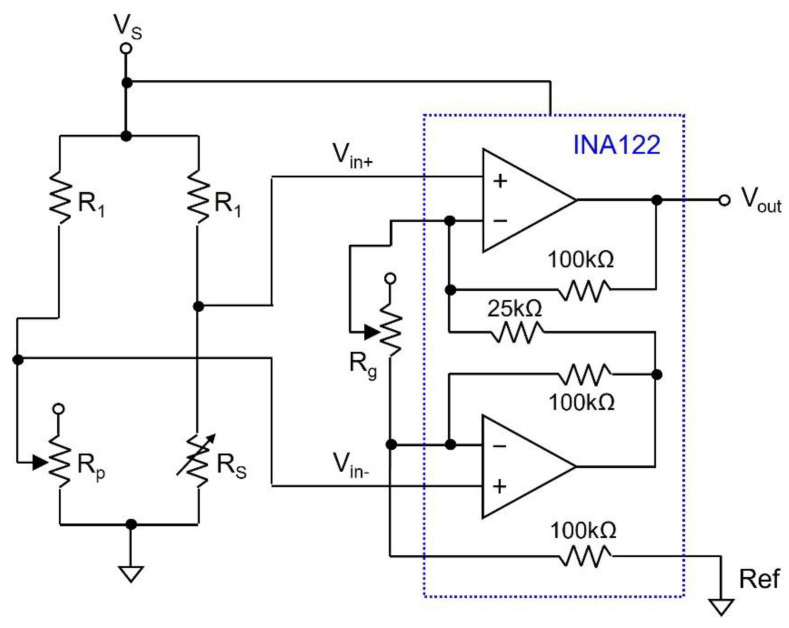
Temperature sensor readout circuit.

**Figure 7 sensors-24-06567-f007:**
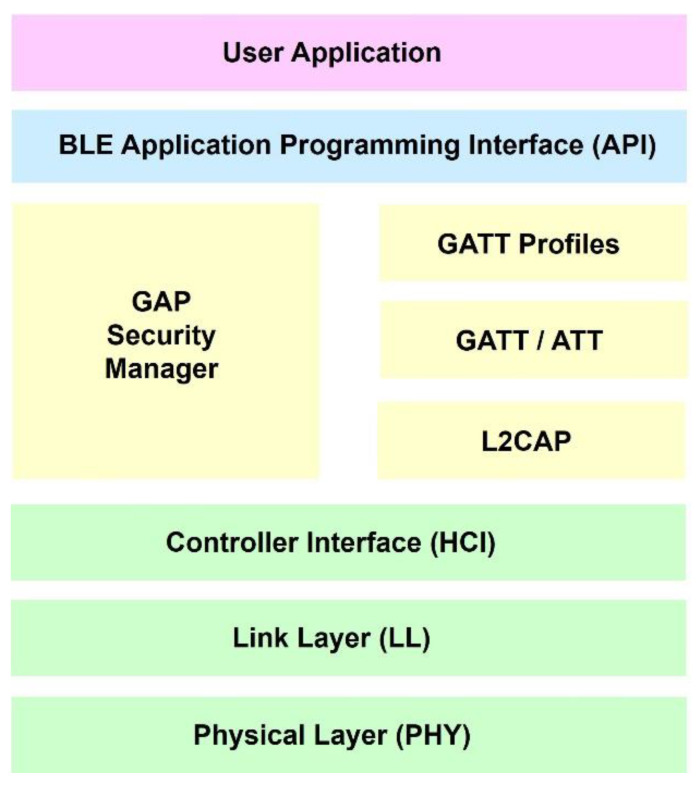
Sensor platform software block diagram.

**Figure 8 sensors-24-06567-f008:**
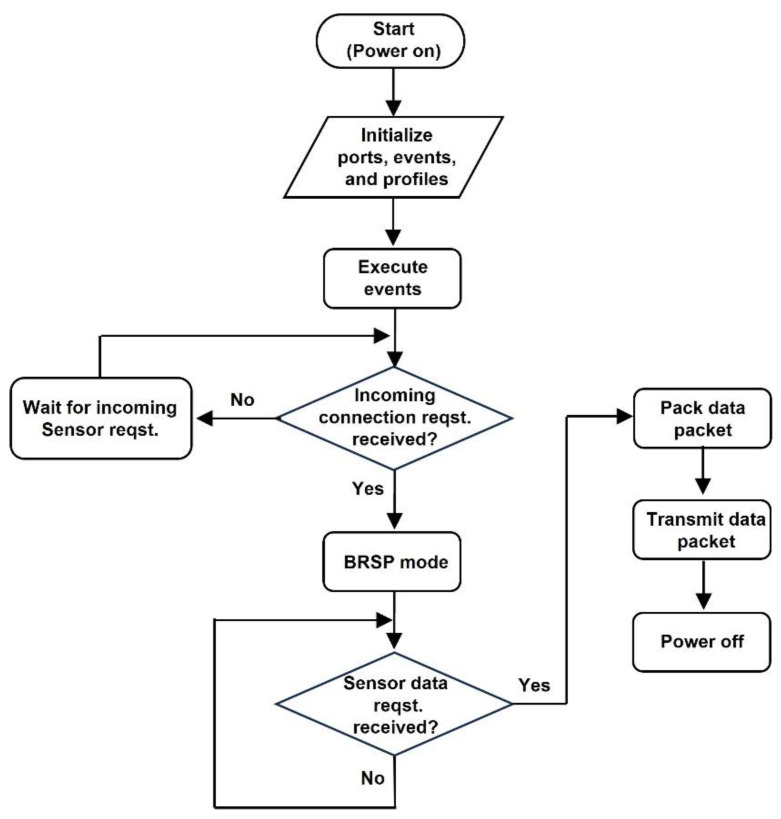
Embedded code flowchart.

**Figure 9 sensors-24-06567-f009:**
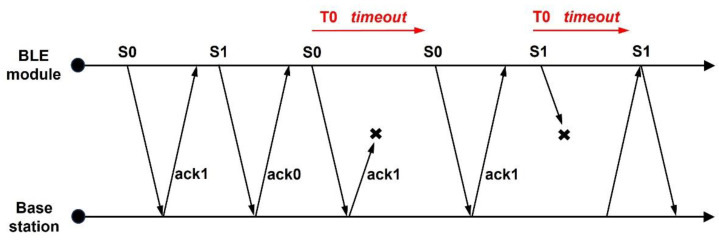
Concept of the stop-and-wait protocol.

**Figure 10 sensors-24-06567-f010:**
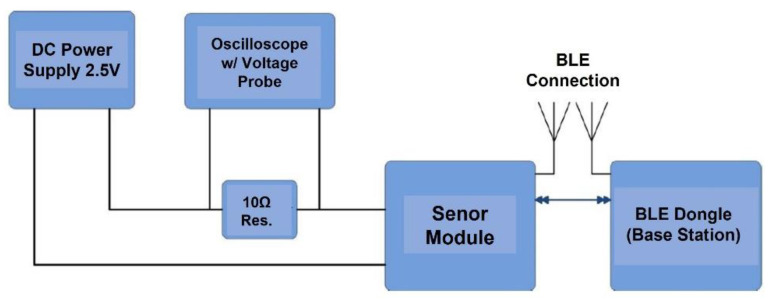
Setup for measuring sensor platform power consumption.

**Figure 11 sensors-24-06567-f011:**
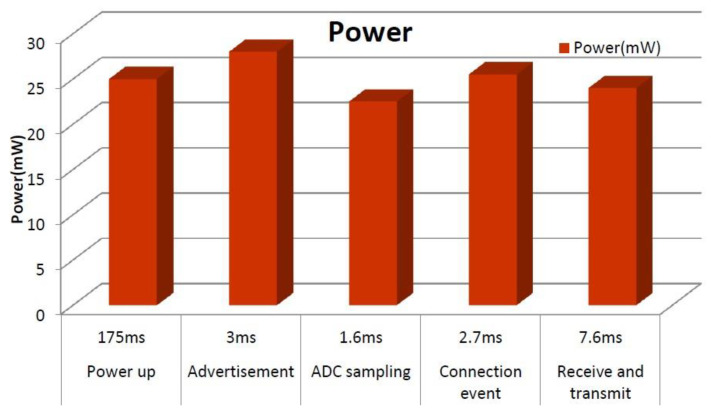
Measured power consumption of the BLE module for each operation stage.

**Figure 12 sensors-24-06567-f012:**
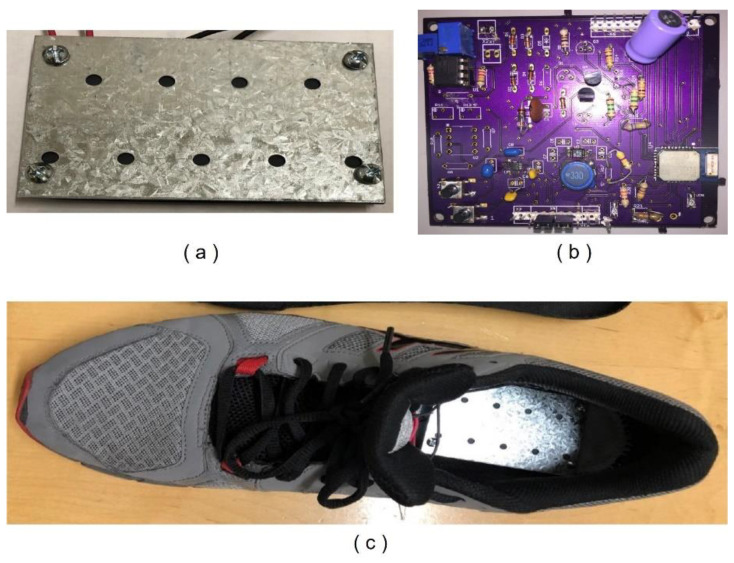
(**a**) Piezoelectric bimorph. (**b**) Sensor platform PCB. (**c**) Bimorph is mounted inside the shoe.

**Figure 13 sensors-24-06567-f013:**
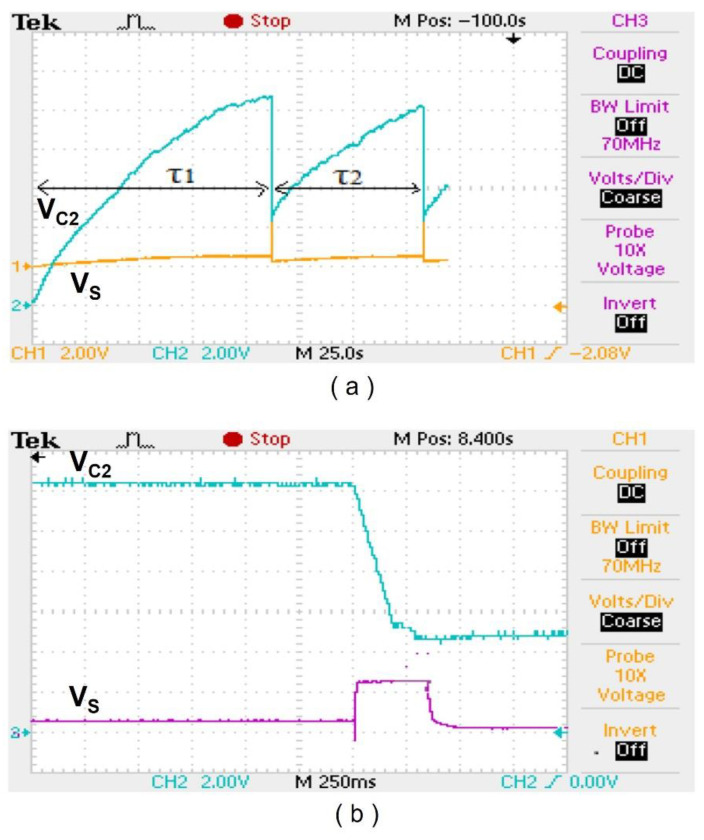
Measured waveform. (**a**) *C*_2_ voltage and regulator output. (**b**) Close-up view of *C*_2_ voltage and regulator output.

**Figure 14 sensors-24-06567-f014:**
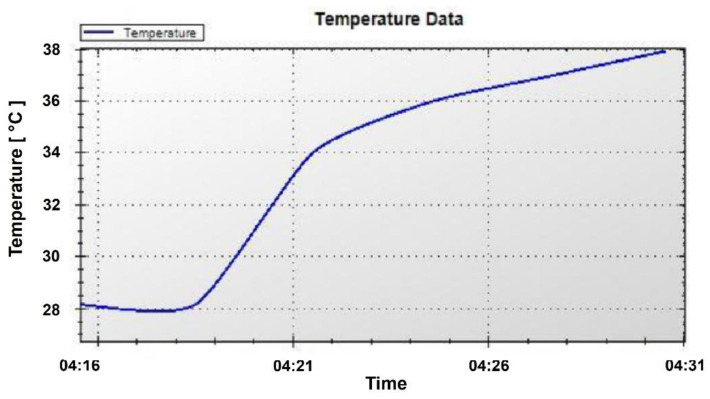
PC application (GUI) for displaying the temperature sensor data.

**Table 1 sensors-24-06567-t001:** Test results of five subjects, operating the proposed foot temperature sensor platform.

Subject No.	1	2	3	4	5
Weight	78.2 kg	85.7 kg	67.4 kg	42.3 kg	56.8 kg
Foot Size	24.6 cm	26.7 cm	23,5 cm	18.2 cm	19.8 cm
*T* _1_	115 s	113 s	120 s	124 s	126 s
*T* _2_	70 s	67 s	72 s	74 s	73 s

**Table 2 sensors-24-06567-t002:** Major features compared to state-of-the-art foot temperature sensors.

Ref	[24]	[25]	[26]	This Work
Sensor type	IR camera	Non-contact IR	LIG	MWCNT
Resolution	-	0.02 °C	0.05 °C	0.1 °C
Wireless module	BLE	Wi-Fi	BLE	BLE
Power source	Battery	Battery	Battery	Piezoelectric bimorph
Shoe mount	No	Yes	Yes	Yes

## Data Availability

Data will be made available upon request.

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
