# Peer review of "A Self-Powered Wireless Temperature Sensor Platform for Foot Ulceration Monitoring"

_sensors, 2024, doi:10.3390/s24206567_

Round 1

Reviewer 1 Report

Comments and Suggestions for Authors

1. The study introduces an innovative solution for monitoring foot ulceration in diabetic patients, which has significant potential in healthcare. However, some areas of the methodology need further clarification. For instance, the study could benefit from additional explanation on how various patient conditions (such as edema, poor circulation, or different foot shapes) were considered during testing. This would strengthen the generalizability of the results across different patient populations.

2. While the study tests the sensor under real-life settings, it would be important to include data on the long-term durability of the piezoelectric bimorph. The article should address how the material behaves after repeated foot movements over a prolonged period, as this directly affects the reliability of the device in real-life applications. The study does not consider the possible long-term degradation of the piezoelectric material, which could affect the sensor’s power generation. This could lead to inaccurate temperature readings over time. It is recommended to include a section discussing the durability of the sensor over months or years. Consider adding an endurance test or projections on the lifespan of the sensor.

3. The device is placed inside the shoe, and it would be helpful to include a section on whether the sensor affects the biomechanics of walking. Patient comfort, gait changes, and potential risks of causing discomfort or injury should be discussed, especially in a population vulnerable to foot issues. A user experience survey or gait analysis would provide valuable insights.

4. There was no clear mention of adjustments for confounding variables such as different shoe types, walking surfaces, or patients' weight. These factors could significantly impact the energy generated and sensor performance. Adding a detailed analysis of how these variables were controlled or adjusted would improve the robustness of the study's findings.

5. The use of Bluetooth Low Energy (BLE) for data transmission raises concerns about data security and patient privacy. Although the focus is on the technical aspects of the sensor, it is crucial to include a section that discusses how data security is maintained. Mentioning the encryption methods or protocols in place to protect sensitive health information would address these concerns.

6. The article does not provide sufficient details on how the sensor performs in varying environmental conditions, such as humidity, temperature, or different foot conditions. These factors are important as they could affect the piezoelectric material’s efficiency or the sensor's accuracy. Including tests or simulations that evaluate the sensor under different environmental conditions would enhance the study's reliability.

7. The study focuses on data collection but does not detail what happens once elevated foot temperatures are detected. Including information on feedback mechanisms (e.g., alerts or instructions for patients) would make the system more practical for preventing ulcerations. A protocol for patients to follow in case of abnormal temperature readings would ensure that the technology leads to actionable outcomes.

8. There is no mention of the cost of producing the sensor platform. Cost is an important factor for medical devices, especially for widespread use among diabetic patients. Including cost analysis could help readers understand its accessibility. An analysis of the potential cost-effectiveness of this platform would be beneficial. Discussing whether the system could reduce long-term healthcare costs for diabetic patients by preventing foot ulcers and amputations would add a valuable perspective. Estimating the return on investment or comparing it to traditional methods of foot monitoring could be insightful for clinical application.

9. The study does not provide enough data on the testing conditions for the diabetic patients. It is important to give more information on the number of patients, how the sensor performed in different foot conditions, and whether the results were consistent. This would make the study more reliable.

10. The sensor calibration process was not described in enough detail. The study mentions the calibration, but it does not explain how it was done or how often the sensor needs to be recalibrated for accurate temperature measurements. This information is essential for future applications.

11. The article does not fully explore the usability of the sensor in different patient lifestyles. For example, how well does the sensor work for patients who are less mobile or have irregular walking patterns? More practical scenarios could improve the study’s applicability.

12. The study mentions that walking is required to generate enough power to send data. However, it does not address situations where patients may be immobile or unable to walk for extended periods. Including alternative methods for data transmission could make the platform more versatile.

13. The paper lacks a thorough risk analysis of potential failure modes. For instance, what happens if the piezoelectric component fails, or if the Bluetooth connection is lost? Addressing these risks would improve the study’s comprehensiveness.

14. The study does not mention a control group for comparison. A control group of patients using a standard foot temperature monitoring system would have provided a better comparison to show the advantages of this new technology.

15. The paper lacks clarity on the statistical methods used to analyze the test results. It would be helpful to include more details on how data was statistically validated to prove the sensor's effectiveness.

16. The paper makes a valuable contribution to the field of diabetic foot monitoring by integrating a self-powered sensor system. The novel use of piezoelectric energy harvesting adds to the growing body of research on sustainable medical technologies. However, additional context about how this system compares to other existing foot monitoring technologies would highlight the unique advantages of the platform.

Reviewer 2 Report

Comments and Suggestions for Authors

The authors demonstrate a self-powered wireless temperature sensor platform that can be used for real-time foot ulceration monitoring for diabetic patients. The work is fascinating and interesting. However, several points are unclear and irrelevant. Below are my major comments that have to be incorporated before the publication in the MDPI journal.

1) Introduction needs a more comprehensive discussion on fundamentals of microheaters, sensor types, temperature monitoring in real time, self powered device, hardware configuration, POCT in medical devices for diagnostics applications. Authors can consider this recent literature: doi.org/10.1016/j.apmt.2024.102225; 10.1016/j.nanoen.2024.109411

2) What is the novelty or significance of this proposed work as there are several reported articles on this proposed topic? Please justify

3) Please tansfer Fig 3, 4, 5 ,6, and 13 to suplementary section as these are poor in resoltuion and does not make any sense

4) Add the table with all the key parameters reported in last 2-3 years of state of the art

5) Whar are the challenges and limitations

6) Please discuss the future direction of this proposed work by providing the schematic

8) Improve the English language

9) Add all the most recent references in this area of research 

10) Please provide the block diagram reresentation of hardwre configuration and software aplication

11) Is this device restricted to ulceration monitory only? Can it be used for other application? Have authors tried using it for some other application?

12) Figure 11 needs labelling 

13) What is the sensitivity and reproduciability rate?

14) Trucate the abstract and rephrase it in a more meaningful way.  

15) Abstract does not provide any summary of the work. Please add in key parameters and gist of the work.  

16) All the figures with graphs should be increased with font, also bold the x axis and y axis for better visualization and enhance the pixel quality  

17) For PMOS and NMOS provide the technical details  

18) Figure 3, power conditioning circuit should be redrawn

19) Figure 4, its hard to under what exactly the authors are explaining 

20) Figure 5, temp readout circuit is not clear and also need explanation

21) Remove figure 6 as it appears to be more noisy and no information

22) Figure 7 flowchart is of very flow resolution please change it

23) Figure 8 needs improvement

24) The technical description in the entire manuscript seems not good and needs rectification and revisions

25) All abbreviation should go in Appendix section

26) Enhance english language

Comments on the Quality of English Language

The authors demonstrate a self-powered wireless temperature sensor platform that can be used for real-time foot ulceration monitoring for diabetic patients. The work is fascinating and interesting. However, several points are unclear and irrelevant. Below are my major comments that have to be incorporated before the publication in the MDPI journal.

1) Introduction needs a more comprehensive discussion on fundamentals of microheaters, sensor types, temperature monitoring in real time, self powered device, hardware configuration, POCT in medical devices for diagnostics applications. Authors can consider this recent literature: doi.org/10.1016/j.apmt.2024.102225; 10.1016/j.nanoen.2024.109411

2) What is the novelty or significance of this proposed work as there are several reported articles on this proposed topic? Please justify

3) Please tansfer Fig 3, 4, 5 ,6, and 13 to suplementary section as these are poor in resoltuion and does not make any sense

4) Add the table with all the key parameters reported in last 2-3 years of state of the art

5) Whar are the challenges and limitations

6) Please discuss the future direction of this proposed work by providing the schematic

8) Improve the English language

9) Add all the most recent references in this area of research 

10) Please provide the block diagram reresentation of hardwre configuration and software aplication

11) Is this device restricted to ulceration monitory only? Can it be used for other application? Have authors tried using it for some other application?

12) Figure 11 needs labelling 

13) What is the sensitivity and reproduciability rate?

14) Trucate the abstract and rephrase it in a more meaningful way.  

15) Abstract does not provide any summary of the work. Please add in key parameters and gist of the work.  

16) All the figures with graphs should be increased with font, also bold the x axis and y axis for better visualization and enhance the pixel quality  

17) For PMOS and NMOS provide the technical details  

18) Figure 3, power conditioning circuit should be redrawn

19) Figure 4, its hard to under what exactly the authors are explaining 

20) Figure 5, temp readout circuit is not clear and also need explanation

21) Remove figure 6 as it appears to be more noisy and no information

22) Figure 7 flowchart is of very flow resolution please change it

23) Figure 8 needs improvement

24) The technical description in the entire manuscript seems not good and needs rectification and revisions

25) All abbreviation should go in Appendix section

26) Enhance english language

Author Response

Plesase, refer to the response letter!

Thank you!

Round 2

Reviewer 1 Report

Comments and Suggestions for Authors

I am happy to tell you that, after careful review, your manuscript is now in very good condition for publication. All of the earlier concerns have been fixed, which has made the paper much better and clearer. Your research is important and adds great value to the field. I appreciate the effort you put into improving the manuscript. I believe the academic community will appreciate your work, and it will help move forward discussions and research in your area.

Reviewer 2 Report

Comments and Suggestions for Authors

Authors have made significant changes to the revised version based on the comments suggested and now looks good and can be accepted for publication in its current form